# Oxygen-rich interface enables reversible stibium stripping/plating chemistry in aqueous alkaline batteries

Haozhe Zhang[1,3], Qiyu Liu[1,3], Dezhou Zheng [2], Fan Yang[1], Xiaoqing Liu[1] & Xihong Lu [1✉]

Aqueous alkaline batteries see bright future in renewable energy storage and utilization, but their practical application is greatly challenged by the unsatisfactory performance of anode materials. Herein, we demonstrate a latent Sb stripping/plating chemistry by constructing an oxygen-rich interface on carbon substrate, thus providing a decent anode candidate. The functional interface effectively lowers the nucleation overpotential of Sb and strengthens the absorption capability of the charge carriers ($SbO_2^-$ ions). These two advantageous properties inhibit the occurrence of side reactions and thus enable highly reversible Sb stripping/plating. Consequently, the Sb anode delivers theoretical-value-close specific capacity (627.1 mA h $g^{-1}$), high depth of discharge (95.0%) and maintains 92.4% coulombic efficiency over 1000 cycles. A robust aqueous $NiCo_2O_4$//Sb device with high energy density and prominent durability is also demonstrated. This work provides a train of thoughts for the development of aqueous alkaline batteries based on Sb chemistry.

[1] MOE of the Key Laboratory of Bioinorganic and Synthetic Chemistry, The Key Lab of Low-carbon Chem & Energy Conservation of Guangdong Province, School of Chemistry, Sun Yat-Sen University, 510275 Guangzhou, People's Republic of China. [2] School of Applied Physics and Materials, Wuyi University, 529020 Jiangmen, People's Republic of China. [3] These authors contributed equally: Haozhe Zhang, Qiyu Liu. ✉email: luxh6@mail.sysu.edu.cn

 1

Demanding requirements for renewable energy storage/utilization greatly stimulate the boom of economical, safe, and efficient batteries[1–5]. With the ever-growing energy consumption, the levelized energy cost (LEC), the economic cost per kW h delivered as output over the entire lifespan of the devices, has becoming the most important indicator of the batteries[6–9]. Therefore, to evaluate the overall performance of the energy storage devices, several parameters need to be taken into account together, including the initial capital cost, specific energy, and cycling ability. Along this line, rechargeable aqueous alkaline batteries (AABs), the devices that realize energy storage via the faradaic reactions of electrodes in alkaline electrolyte, emerge as one of the most promising next-generation candidates for renewable energy storage, especially for large-scale applications[10,11]. This should be ascribed to the advantageous properties of aqueous electrolytes including high ionic conductivity ($\sim 1\ S\ cm^{-1}$), low cost, and intrinsic nonflammability[12]. Over the past years, tremendous research attention has been paid to the exploration of various electrode materials for AABs, and significant achievements have been gained in this field[13–16]. Compared with the booming cathode materials, the choice of anode materials remains quite limited and their development is relatively slow.

Currently, based on the disparate differences of their energy storage behavior, the reported anodes are mainly classified into two categories: conversion-type ones and stripping/plating-type ones. Conversion-type anodes such as Cd[17], Bi[18,19], and $FeO_x$[20,21] experience phase conversion (usually from metal/metal oxide to their corresponding metal oxide/hydroxide) during the charging–discharging process. Yet, the intrinsically poor conductivity and slow reaction kinetics of the metal oxides/hydroxides lead to low capacity and limited rate capability. In contrast, Zn anodes, the most representative stripping/plating-type electrodes, are not bothered by such troubles because they rely on the transformation of metal/metal ions for energy storage, which ensures fast reaction kinetics[22–24]. Unfortunately, due to the uncontrollable dendrite growth of Zn and the accompanying side reactions, this metal anode suffers from inferior cycling stability and low coulombic efficiency (CE), seriously impeding its further development[25]. Therefore, it is of great significance to explore novel anode materials embedding both high energy and favorable stability to facilitate the practical applications of AABs in renewable energy storage/utilization, which remains a great challenge in this area.

Stibium (Sb) metal is a desirable anode material owing to its high theoretic capacity ($660\ mA\ h\ g^{-1}$, based on three-electron transfer reaction), favorable negative redox potential in alkaline solution ($-0.66\ V$ vs. the standard hydrogen electrode) and low cost ($\sim 7\$$ per kg)[26,27]. Furthermore, according to the φ-pH diagram for $H_2O$–Sb and previous studies, the Sb (III) will exist as soluble $SbO_2^-$ rather than $Sb_2O_3$ precipitate in aqueous alkaline environment, which endows Sb metal the capability to function more like the stripping/plating-type anodes with fast kinetics in AABs[28–31]. To our knowledge, there is hardly any report on the employment of metallic Sb as electrode in aqueous systems, let alone in AABs. For a Sb-based AAB device, there exists strong electrostatic repulsion force between the charge carriers ($SbO_2^-$) and the electrode surface, both negatively charged during the plating courses. Such specific interaction makes it difficult for $SbO_2^-$ ions to approach the anode for proper deposition, and might initiate severe side reactions such as hydrogen evolution. Hence, the key challenge lies in how to manipulate precisely the behaviors of $SbO_2^-$ ions at the interface.

Herein, by constructing a functional oxygen-rich interface, we realize highly reversible stripping/plating chemistry of Sb metal anode on the carbon substrate (denoted as CS) in AABs. Oxygen-containing functional groups facilitate the diffusion and deposition behaviors of $SbO_2^-$ ions on the carbon surface via two ways: (i) to promote the absorption of the $SbO_2^-$ at the interface by activating the formation of hydrogen bonds; (ii) to decrease the deposition resistance of Sb by minimizing the nucleation overpotential on the anode. Benefiting from these two merits, the potential side reactions are substantially inhibited and highly reversible deposition/dissolution of Sb is realized on the functionalized carbon substrate (denoted as FCS). As a result, the Sb/FCS anode delivers a high specific capacity of $627.1\ mA\ h\ g^{-1}$ (~95.0% depth of discharge, DOD), along with admirable CE (~95.9%) and satisfactory cycling durability (92.4% CE after 1000 cycles). When coupled with a phosphating $NiCo_2O_4$ (denoted as P-$NiCo_2O_4$) cathode, the electrochemical performance of the AAB device outperforms most recently reported AABs, as testified by its superior energy density ($8.2\ mW\ h\ cm^{-3}$), excellent power density ($0.4\ W\ cm^{-3}$), as well as its admirable stability (98.1% capacity retention over 1000 cycles). This work opens a favorable way for the exploration of novel Sb-based aqueous devices as power supply systems.

## Results

**Stripping/plating chemistry studies of stibium.** The stripping/plating chemistry of Sb on different substrates is compared in Fig. 1a. Because the Sb (III) will exist as $SbO_2^-$ without forming tartrate-Sb complex in potassium antimony tartrate-KOH aqueous system, the simplified schematic illustration only shows the $K^+$ and $SbO_2^-$ for demonstration[28]. For the CS substrate, at the very beginning, the disordered molecular thermodynamic motion of the $SbO_2^-$ ions leads to their random distribution in the electrolyte. Upon charging the anode, the $SbO_2^-$ ions in the Helmholtz layer are reduced to Sb on the substrate, and the as-deposited Sb metal would resolve into the electrolyte during the discharging course. It is noteworthy that, in the charging process, a large portion of $SbO_2^-$ ions would migrate toward the counter electrode instead of the deposition substrate, leading to catastrophic plating efficiency or even side reactions. This should be attributed to the strong electrostatic repulsion between the charge carriers ($SbO_2^-$) and the negatively-charged CS surface. Therefore, to enable highly reversible stripping/plating chemistry of Sb, it is essential to design a functionalized carbon interface that is capable of "capturing" tightly the $SbO_2^-$ in the close vicinity of the electrode at the deposition stage. To achieve this goal, we decorate the CS with some oxygen-containing functional groups via a facile electrochemical activation strategy. As shown in Supplementary Fig. 1, the untreated CS consisting of interlaced carbon fibers exhibits a smooth surface and it becomes relatively rough after electrochemical treatment. The Brunauer–Emmett–Teller surface areas of both samples shown in Supplementary Fig. 2 remain small values, but which of FCS becomes slightly larger after the treatment (from 1.4 to $3.2\ m^2\ g^{-1}$). The introduction of oxygen functional groups in FCS is accompanied by the generation of graphene edges with thickness of about 10 nm (Supplementary Fig. 3). In addition, the X-ray diffraction (XRD) spectra of CS and FCS (Supplementary Fig. 4) are both perfectly indexed to hexagonal graphite (JCPDF#41-1487), indicating their similar crystalline structure[32]. Yet, the detailed X-ray photoelectron spectroscopy (XPS) analysis of the C 1s peak reveals that the electrochemical activation process successfully introduces oxygen atoms to the FCS surface in the form of C–OH (285.7 eV), C=O (286.9 eV), and O–C=O (288.7 eV) (Supplementary Fig. 5)[33]. Congruously, the intensity of O 1s peak for the FCS (O/C ratio = 0.17) is also much higher than the CS (O/C ratio = 0.04). Electrochemical impedance spectra (EIS) in Supplementary Fig. 6 indicates that the charge transfer resistance of FCS is slightly increased, consistent with the previous work[32]. Nevertheless, the

oxygen-rich interface holds great potential for boosting the Sb deposition because it can provide abundant receptors and donors for the formation of hydrogen bonds that favors the adsorption of $SbO_2^-$ ions in the Helmholtz layer.

The working potential window of the CS and FCS electrodes in 1 M KOH with 0.027 M $C_8H_4K_2O_{12}Sb_2$ was tested by linear sweep voltammetry (LSV) at 1 mV s$^{-1}$. As shown in Supplementary Fig. 7, the H$_2$ evolution reactions on both two electrodes take place at around −1.5 V, so the potential window of 0 to −1.3 V was selected to avoid water splitting. Cyclic voltammograms (CV) within this voltage window were recorded on the CS and FCS at 2 and 10 mV s$^{-1}$ to study the stripping/plating chemistry of Sb (Supplementary Fig. 8). Both electrodes possess one redox couple whereas smaller voltage polarization is visualized on the FCS sample, indicating its higher reversibility. Further chemical composition investigation verifies the redox couple corresponds to the deposition/dissolution of Sb (Fig. 1b–g and Supplementary Fig. 9). Specifically, when the charging capacity is set to 0.47 mA h cm$^{-2}$, the precipitates on both electrodes are confirmed to be pure hexagonal R-3m phase Sb without other impurities (JCPDS#85-1322, Supplementary Fig. 10)[34,35]. The high-resolution transmission electron microscopy (HRTEM) characterization in Supplementary Fig. 11 further testifies this viewpoint. The lattice spacing of 0.31 nm matches well with the (012) plane of hexagonal Sb (JCPDS#85-1322), while the bright diffraction and highly-ordered spots in the selected-area electron diffraction (SAED) patterns indicate its good crystallinity[36]. When the electrodes are discharged to 0 V (discharging state), Sb metal are nearly completely dissolved back to the electrolyte. Notably, in the charging stage, the Sb metal (Fig. 1d) deposits as randomly-distributed, isolated nanoflowers, leaving some uncovered spaces on the CS. In contrast, the deposition of Sb seems more homogeneous on the entire surface of the FCS (Fig. 1e), indicative of the superiority of the oxygen-decorated functional interface. The Sb/$SbO_2^-$ transformation during the charging/discharging is also verified by the *ex situ*

XRD spectra (Supplementary Fig. 12), in which the characteristic peaks of metallic Sb disappear at the discharging state and recover at the charging state.

To highlight the superiority of the FCS over the CS, their electrochemical behaviors are compared with a fixed charging capacity of 0.47 mA h cm$^{-2}$. At a current density of 30 mA cm$^{-2}$, the charge limited voltage ($V_{cl}$) of Sb/CS, which means the voltage attained at the end of charging (absolute value), reaches a high value of 1.50 V within merely 50 cycles and increases to 1.70 V after 500 cycles, accompanied by severe hydrogen evolution (Supplementary Fig. 13). The Sb/FCS electrode shows a stable charging voltage profile, along with a low $V_{cl}$ about 1.25 V for 1000 cycles (Fig. 2a). At a smaller current density of 20 mA cm$^{-2}$ (Fig. 2b), the hydrogen evolution of the Sb/CS electrode is intensified, resulting in a high $V_{cl}$ of 1.55 V after only ten cycles. In contrast, the $V_{cl}$ of the Sb/FCS electrode is only 1.30 V after 1000 cycles (Supplementary Fig. 14), signifying its better cycling durability. Moreover, it should be noticed that the test duration of Sb/FCS at both current densities lasts much longer than Sb/CS. The charging time and cycling numbers of Sb/CS and Sb/FCS are the same, so the longer test time means longer average discharging time, manifesting longer average discharging time and better CE of Sb/FCS.

According to the discharging curves at 20 mA cm$^{-2}$ in Fig. 2c, the Sb/CS show a capacity of ~0.32 mA h cm$^{-2}$ at the first 50 cycles. When the cycling test is extended from 100 to 1000 cycles, it fades rapidly from 0.18 to 0.05 mA h cm$^{-2}$, signifying the irreversible plating/stripping courses. Obviously, the Sb/FCS maintains a much larger discharging capacity in the range of 0.42–0.45 mA h cm$^{-2}$ during the 1000 cycles. At lower current density conditions (2–8 mA cm$^{-2}$), the discharging capacities of Sb/FCS can even achieve more than 0.46 mA h cm$^{-2}$ (Supplementary Fig. 15). It is worth noting that the capacitive charge contribution of both CS and FCS only account for ignorable percentage comparing with the discharging capacity of Sb/CS (~1.3%) and Sb/FCS (~2.8%), indicating the capacities are mainly

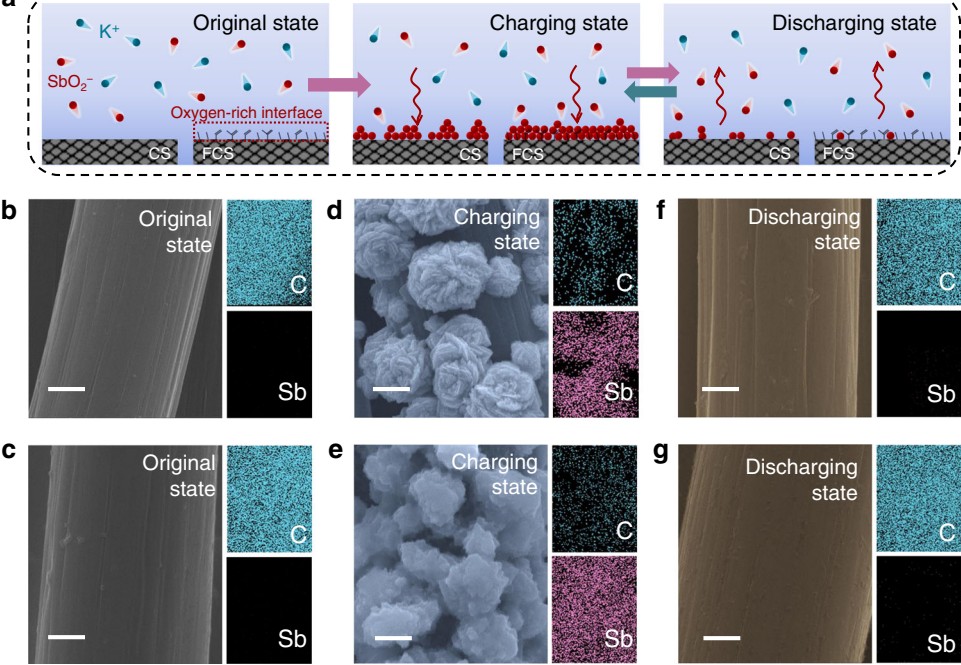

**Fig. 1 Charge storage mechanism investigations of stibium. a** Schematic illustration of the charge storage mechanism of the Sb anode in 1 M KOH and 0.027 M $C_8H_4K_2O_{12}Sb_2$ electrolyte. The arrows indicate the direction of Sb metal stripping/plating. **b–g** SEM images and mapping of **b** CS and **c** FCS in original state; **d** Sb/CS and **e** Sb/FCS in charging state; **f** Sb/CS and **g** Sb/FCS in discharging state. Scale bars: 2 μm.

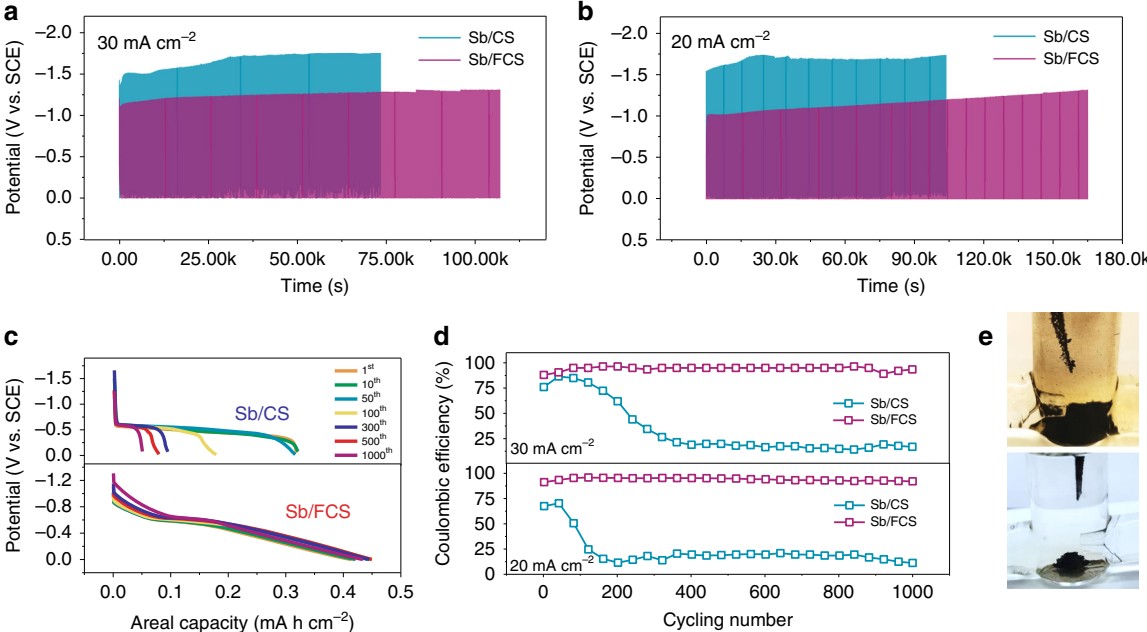

**Fig. 2 Electrochemical characterizations of Sb/CS and Sb/FCS electrodes.** Voltage profiles of Sb/CS and Sb/FCS with a fixed charging capacity of 0.47 mA h cm$^{-2}$ at **a** 30 mA cm$^{-2}$ (55 s charging) and **b** 20 mA cm$^{-2}$ (85 s charging). **c** Corresponding discharging curves at different cycles at 20 mA cm$^{-2}$ and **d** CE of the Sb stripping/plating on the bare CS and FCS. **e** Optical photographs of Sb/CS and Sb/FCS after 1000 cycles at 20 mA cm$^{-2}$.

come from the stripping/plating process of Sb (Supplementary Fig. 16). Additionally, an impressive DOD of 95% and a superb specific capacity of 627.1 mA h g$^{-1}$ are achieved by the Sb/FCS anode (based on the mass loading of Sb), outstripping various anodes of aqueous batteries. Besides, the CE (the ratio of Sb stripping capacity to Sb plating capacity) of the two samples are also presented in Fig. 2d to elucidate the sustainability of the electrodes. As expected, at 20 or 30 mA cm$^{-2}$, the CE of FCS electrode remains to be 92–95% without attenuation after 1000 cycles. In comparison, the CE of the CS electrode drops to less than 20% very shortly at both current densities. Such disappointed performance might be caused by the occurrence of severe side reactions. The morphology characterization of the electrodes after cycling tests shows that the deposition/dissolution processes of Sb on the CS are irreversible, leaving much more black sediments (insoluble "dead Sb") (Fig. 2e)[37]. Yet, no obvious difference is seen on the FCS before and after the stability test (Supplementary Fig. 17). It is thus comprehensively demonstrated that the oxygen-rich interface of FCS could restrict the side reactions and guide the reversible stripping/plating chemistry of Sb.

**Insights into reversible stripping/plating process.** The role of the oxygen-rich interface of FCS in inducing reversible Sb stripping/plating chemistry was then investigated theoretically by density functional theory (DFT) calculations. During the plating process, the absorption properties of SbO$_2^-$ on the substrate surface have a vital effect on the Sb/SbO$_2^-$ conversion reaction. Figure 3a describes the absorption energy of the SbO$_2^-$ for the CS surface with or without the introduction of the oxygen-containing functional groups. Remarkably, the surfaces with all kinds of functional groups including >C=O ($-2.0$ eV), >C–OH ($-1.6$ eV) and −COOH ($-2.1$ eV) possess much lower absorption energy than that of the bare CS ($-0.9$ eV). Therefore, the FCS surface is very likely to gather more SbO$_2^-$ ions in the Helmholtz layer, which would significantly enhance the Sb plating efficiency. As a proof of concept, when fixing the charging capacity at 0.47 mA h cm$^{-2}$, the electrochemical reactions on CS can be divided into

two stages (Fig. 3b): (i) at around $-1.0$ V, an efficient Sb deposition accounts for 0.17 mA h cm$^{-2}$; (ii) with the rapid potential increase to $-1.6$ V, the hydrogen evolution side reaction (HER) makes up the rest capacity of 0.30 mA h cm$^{-2}$. This point can be further proved by the weight variation of CS electrode during the charging process (Supplementary Fig. 18). As illustrated in Fig. 3c, d, when the CS is negatively charged for Sb plating, the strong electrostatic repulsion force repels the SbO$_2^-$ ions to diffuse away from the anode surface. When the few SbO$_2^-$ ions left in the Helmholtz layer is completely consumed by initial Sb deposition, water decomposition reaction will be launched alternatively due to the lack of Sb source, leading to serious hydrogen evolution and active material shedding. Benefitting from the oxygen-rich functional interface, the supplementary of SbO$_2^-$ in the Helmholtz layer would be substantially more effective on the FCS during the plating. Under this circumstance, the FCS enables an efficient Sb deposition up to 0.47 mA h cm$^{-2}$ at a low cut-off charge voltage of $-1.0$ V (Fig. 3b), getting rid of the interference of HER and thus achieving a high CE. This is highly consistent with the XPS C 1$s$ spectra at the charging state in Supplementary Fig. 19, in which the Sb/FCS holds a raised quantity of C–Sb bonds (C–Sb: C–C = 0.61–0.27) compares with the Sb/CS[38]. Moreover, the deposition of Sb on FCS can be devided into two stages: (i) 0 to $-0.95$ V and (ii) $-0.95$ to $-1.0$ V. As revealed by the weight variation curve and *ex situ* scanning electron microscopy (SEM) images of the charging process in Supplementary Fig. 20, the electrochemical deposition of Sb take place on FCS at both stages. The first stage (0 to $-0.95$ V) might be attributed in an underpotential deposition (UPD)-like process, while the other stage ($-0.95$ to $-1.0$ V) is normal deposition process. The nucleation stage is a pivotal step for fully understanding the deposition behaviors of Sb on different substrates[39]. We then compare the nucleation overpotential disparities of CS and FCS to illuminate the role of oxygen-rich functional interface in modulating Sb nucleation. The nucleation overpotential is defined as the voltage difference between the lowest voltage, when sharp voltage drops take place and equilibrium potential in the galvanostatic cathodic polarization. As shown in Supplementary

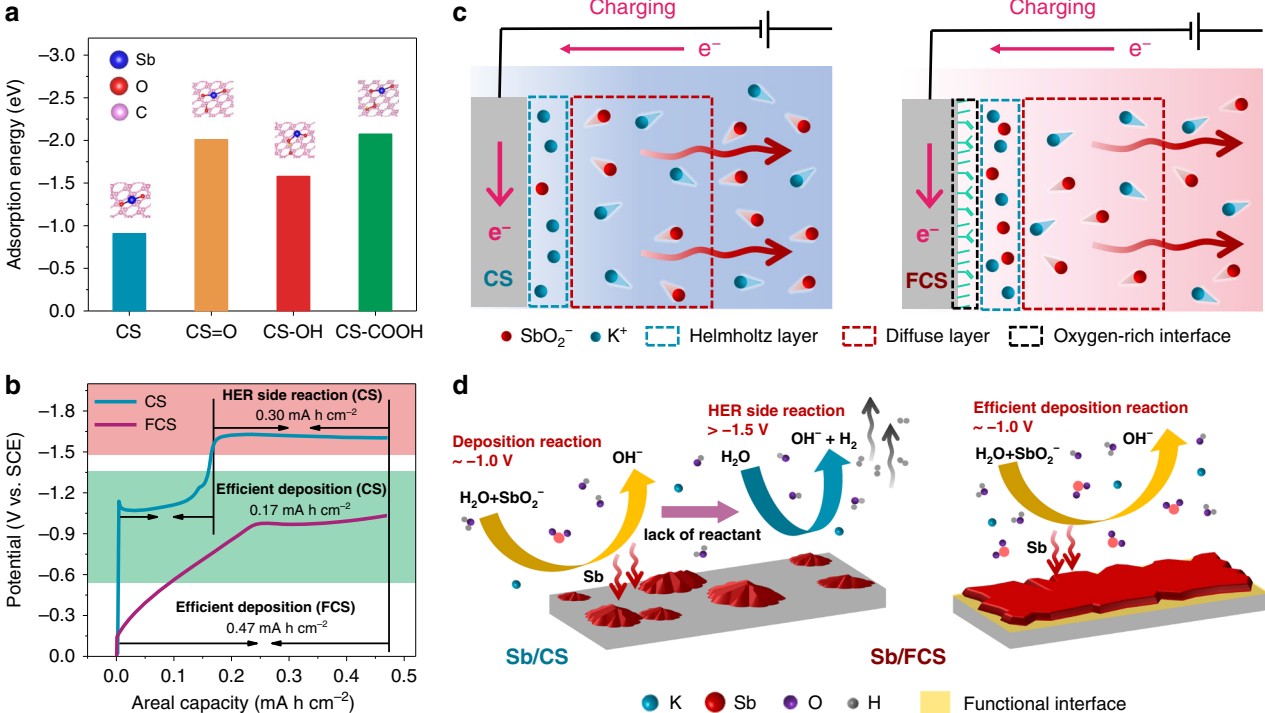

**Fig. 3 Comprehension of oxygen-rich interface induced reversible stripping/plating process. a** Relative $SbO_2^-$ absorption energy profiles of the CS surface with different functional groups. **b** Electroplating curve of the Sb on the substrates at 20 mA cm$^{-2}$ with a fixed charging capacity of 0.47 mA h cm$^{-2}$. **c** Illustration of the ion distribution during the plating process of CS and FCS. The arrows indicate the direction of $SbO_2^-$ diffusion. **d** The plating mechanism on the substrate with/without the functional interface induced.

Fig. 21, with respect to the CS, the profile of FCS delivers a smoother voltage dip and a much lower nucleation overpotential (from 147 to 29 mV), suggesting the oxygen-rich functional interface can dramatically lower the resistance for Sb nucleation and is more favorable for Sb deposition. Briefly, the FCS integrates the strong absorption capability of the $SbO_2^-$ and the lower Sb nucleation overpotential, which conjointly contribute to the suppression of side reactions and the high efficiency of Sb stripping/plating.

**Process performance of NiCo//Sb aqueous alkaline battery.** In order to further demonstrate the potential of Sb/FCS electrode in energy storage applications, we assembled an AAB device (denote as NiCo//Sb battery) by employing Sb/FCS as anode and P-NiCo$_2$O$_4$ (see details in Supplementary Fig. 22 and see "Methods" section) as cathode[40]. Figure 4a shows the CV curves of the cathode and the anode at 10 mV s$^{-1}$. The large potential difference between the two electrodes brings about a high discharging platform of ~1.0 V, which is consistent with the CV results of NiCo//Sb battery (Supplementary Fig. 23). The energy storage mechanism of the NiCo//Sb battery is illustrated in Fig. 4b, in which the cathode proceeds redox reactions while the anode undergoes stripping/plating process (Supplementary Note 1)[22,31,41].

It is noteworthy that the as-fabricated battery delivers a large areal capacity of 0.75 mA h cm$^{-2}$ at 8 mA cm$^{-2}$ (Fig. 4c). With the increase of the current density to an ultrahigh level of 36 mA cm$^{-2}$, 0.21 mA h cm$^{-2}$ capacity is still retained with only 21 s discharging, reflecting its good rate ability. When the active material mass of both electrodes is counted, the NiCo//Sb AAB also reaches a high specific capacity of 175.6 mA h g$^{-1}$, superior to recently reported AABs shown in Fig. 4d[19–21,42,43]. Furthermore, a maximum volumetric energy density of 8.2 mW h cm$^{-3}$ and a maximum volumetric power density of 0.4 W cm$^{-3}$ have been achieved of the

battery (based on combined volume of cathode and anode), which considerably surpass a series of AABs (Fig. 4e), like Ni–NiO//BiO$_x$ battery (1.6 mW h cm$^{-3}$, 0.44 W cm$^{-3}$)[18], NiCo$_2$O$_4$//Bi battery (1.5 mW h cm$^{-3}$, 0.02 W cm$^{-3}$)[19], NCHO (nickel cobalt hydroxide)//Zn battery (2.2 mW h cm$^{-3}$, 0.05 W cm$^{-3}$)[44], Ni(OH)$_2$//FeO$_x$ battery (2.5 mW h cm$^{-3}$, 0.03 W cm$^{-3}$)[20], Ni/N-doped C//Zn battery (0.66 mW h cm$^{-3}$, 0.13 W cm$^{-3}$)[45], NiO//Fe$_3$O$_4$ battery (5.2 mW h cm$^{-3}$, 0.08 W cm$^{-3}$)[21], Ni/Ni(OH)$_2$//Zn battery (2.9 mW h cm$^{-3}$, 0.14 W cm$^{-3}$)[46]. More encouragingly, our NiCo//Sb battery also exhibits outstanding long-term cycling stability with a slight capacity fading (~1.9%) after 1000 cycles (Fig. 4f), as well as a high CE (95.7%). The charge/discharge profiles of the battery at the initial, middle and final stages remain almost the same in the voltage plateaus and charging/discharging times, again demonstrating its remarkable reversibility and stability.

**Discussion**

In summary, we demonstrated a highly reversible stripping/plating chemistry of Sb and thus provide a promising electrolytic metallic Sb anode for AABs. The diffusion behaviors of $SbO_2^-$ ions at the interface are effectively manipulated by an oxygen-rich functional interface, which could not only significantly promote the surface absorption of $SbO_2^-$ in the Helmholtz layer to restrain the side reaction and increase the CE; but also, lower the nucleation overpotential and induce the uniform Sb deposition. As a consequence, the Sb/FCS anode delivered an obviously improved durability with an excellent CE of more than 92.4% during the 1000 cycles, associated with a remarkable specific capacity of 627.1 mA h g$^{-1}$ at 0.47 mA h cm$^{-2}$ (~95.0% DOD). Moreover, the as-prepared NiCo//Sb AAB device delivered a prominent stability (98.1% capacity retention after 1000 cycles), an outstanding maximum volumetric energy density (8.2 mW h cm$^{-3}$) along with a peak power density (0.4 W cm$^{-3}$), surpassing many of the state-of-the-art AABs reported recently. The development of the

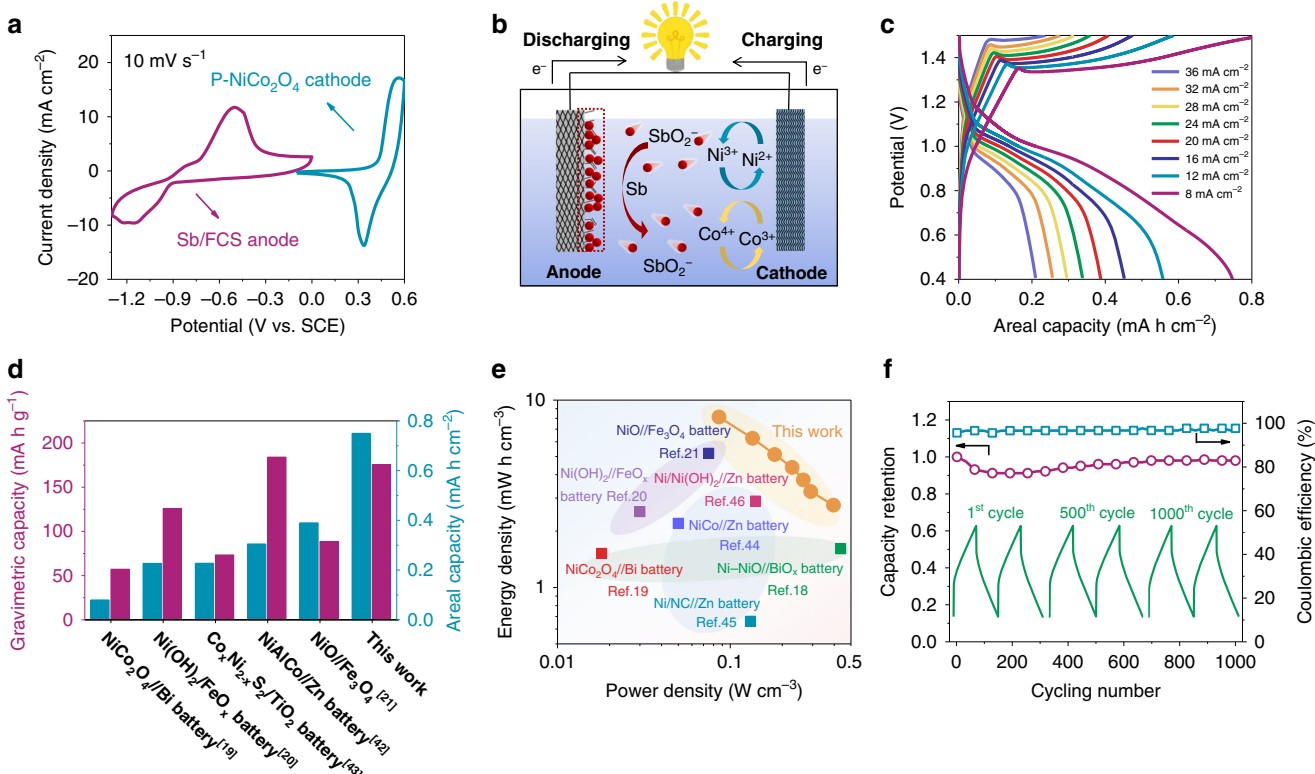

**Fig. 4 Electrochemical performance of assembled AAB using Sb/FCS anode. a** CV curves of the P-NiCo$_2$O$_4$ cathode and Sb/FCS anode measured at 10 mV s$^{-1}$. **b** The energy storage mechanism of the NiCo//Sb battery. **c** Galvanostatic charge–discharge (GCD) curves at different current densities, **d** areal capacity and gravimetric capacity comparison with previous studies[19–21,42,43], **e** Ragone plots, the values reported for other energy storage devices are added for comparison[18–21,44–46] and **f** the cycling stability of the NiCo//Sb battery.

promising Sb anode in our work will be of immediate benefit to the exploration of high-performance AABs for practical utilization, particularly, for grid-scale energy storage.

## Methods

**Preparation of electrolyte**. All reagents are of analytical grade and are directly used without any purification. 0.5 g of potassium antimony tartrate (C$_8$H$_4$K$_2$O$_{12}$Sb$_2$) is dissolved in 30 ml 1 M KOH, resulting in a mixture of 1 M KOH and 0.027 M C$_8$H$_4$K$_2$O$_{12}$Sb$_2$. The reaction equation for the formation of SbO$_2^-$ is listed in Supplementary Note 2.

**Preparation of CS and FCS electrodes**. The CS substrate is purchased directly from Fuel Cell Earth, USA (plain carbon fiber cloth; 12.6 mg cm$^{-2}$). Subsequently, the FCS substrate is obtained by CS after an electrochemical functional group introducing process. First, the CS is ultrasonic in water and ethanol for 10 min and undergoing controllable Ar plasma process by atomic layer deposition (ALD) technology (ALD-SC6-PE, Syskey Technology Co., Ltd.) to produce clean surface. The electrochemical functional group introducing is then carried out through a standard three-electrode system in a solution composed by H$_2$SO$_4$ (98%) and HNO$_3$ (68%) with a volume ratio of 1:1. The system uses Pt electrode as the counter electrode, saturated calomel electrode (SCE) as the reference electrode and CS as the working electrode. The electrochemical functional group introducing process is conducted with a constant voltage of 3 V for 15 min at the room temperature. The FCS is washed with deionized water after the process and dried in the oven at 60 °C for 3 h.

**Preparation of P-NiCo$_2$O$_4$ cathode**. P-NiCo$_2$O$_4$ is obtained by a previously reported hydrothermal and annealing method growing on nickel foam. Initially, urea (15 mmol), NiCl$_2$·6H$_2$O (5 mmol), and CoCl$_2$·6H$_2$O (10 mmol) are dissolved in 75 ml deionized water under vigorous magnetic stirring. After the mixture is clarified, 35 ml solution is added into a 50 ml Teflon-lined stainless steel autoclave with a piece of cleaned nickel foam (2.7 cm × 4 cm). The autoclave is heated at 120 °C for 6 h and then cooled down to room temperature. After washed with deionized water and dried at 70 °C in air, the prepared NiCo$_2$O$_4$ is calcined at 300 °C for 2 h and the heating rate was 2 °C min$^{-1}$. To get P-NiCo$_2$O$_4$, the obtained samples are annealed at 250 °C for

60 min in Ar atmosphere in the presence of NaH$_2$PO$_2$·H$_2$O (1.2 g). The mass loading of the P-NiCo$_2$O$_4$ is 3.1 mg cm$^{-2}$ (BT25S, 0.01 mg), and the thickness of the cathode after tableting is 0.04 cm.

**Electrochemical measurements**. Galvanostatic charge/discharge curves (GCD) and cyclic voltammogram (CV) are recorded using Neware battery system (CT-3008-5V10mA-164, Shenzhen, China) and electrochemical work-station (CHI 760E). The Princeton electrochemical workstation (PARSTAT MC) was used to collect Electrochemical impedance spectroscopy (EIS). All electrochemical characterization is performed at room temperature. For Sb plating/stripping test, CS and FCS substrates are used as working electrode with a surface area of 0.5 cm$^2$, graphite rod was employed as counter electrode, and SCE as the reference electrode in three-electrode system. The electrolyte is a mixed solution of 1 M KOH and 0.027 M C$_8$H$_4$K$_2$O$_{12}$Sb$_2$. The charging process is achieved by a galvanostatic charging method to charge a fixed value (0.47 mA h cm$^{-2}$) and the discharging process is achieved by a galvanostatic discharging method to discharge to 0 V. The mass loadings of Sb/CS and Sb/FCS deposition electrodes are 0.64 mg cm$^{-2}$ and 0.71 mg cm$^{-2}$, respectively while charging 0.47 mA h cm$^{-2}$ (BT25S, 0.01 mg). The aqueous NiCo//Sb battery is tested in two-electrode system with the same electrolyte, using P-NiCo$_2$O$_4$ as cathode and FCS substrate as anode. The mass loading of Sb/FCS electrode at 8 mA cm$^{-2}$ is 1.1 mg cm$^{-2}$. Calculations about capacity, energy density and power density of Sb anode and NiCo//Sb battery are shown in Supplementary Note 3.

**Material characterization**. Field-emission SEM (SEM, JSM-6330F and SEM, g-500) and TEM (FEI Tecnai G$^2$ F30) are used to character the morphology and the microstructure of the deposited Sb on the substrate. XRD (D-MAX 2200 VPC, RIGAKU), XPS (NEXSA, Thermo FS) are used to character the crystal phase and composition of the deposited Sb on the substrate.

**Computational details**. All the calculations were performed using the Vienna Ab-initio Simulation Package (VASP). The generalized gradient approximation (GGA) in the scheme of Perdew–Burke–Ernzerhof (PBE) function was used to calculate the electron exchange-correlation interactions. The cutoff energy for plane-wave basis set was set to 400 eV. All atomic positions and lattice vectors were fully optimized using a conjugate gradient algorithm to obtain the unstrained

configuration. Atomic relaxation was performed until the change of total energy was less than $1 \times 10^{-5}$ eV, all the forces on each atom were smaller than 0.01 eV/Å. The adsorbed surface was used (002) slab with 12 Å vacuum in the c-direction to eliminate periodic boundary interaction. Monkhorst–Pack scheme K-point grid was set to $3 \times 3 \times 1$ during surface relaxation.

## Data availability

The data that support the findings of this study are available from the corresponding authors upon reasonable request.

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

## Acknowledgements

We thank the financial support by the National Natural Science Foundation of China (21822509, U1810110, and 21802173), Science and Technology Planning Project of Guangdong Province (2018A050506028), and Natural Science Foundation of Guangdong Province (2018A030310301). The calculation of this work was performed on TianHe-2, thanks for the support of National Supercomputing Center in Guangzhou (NSCC-GZ). The authors also thank the Photoemission Endstations (BL10B) in National Synchrotron Radiation Laboratory (NSRL) for help in characterizations.

## Author contributions

X.Lu and H.Z. planned and designed the project. H.Z. and Q.L. fabricated the materials and performed the electrochemical experiments. D.Z. performed and analyzed the XPS result.

F.Y. conducted the DFT analysis. X.Lu, H.Z., Q.L., and X.Liu analyzed the data and wrote the manuscript. All authors discussed the results and commented on the manuscript.

## Competing interests

The authors declare no competing interests.
