## [Peer Review File · Nature Communications]

Reviewer #1 (Remarks to the Author):

Authors reported the Sb-based charge storage system, which involves the plating/stripping reaction similar to Zn based system. The work is interesting in the academic aspect and certainly deserves the space. However, the obtained results are not in the level of Nature Communications. Therefore, I am not recommending it for publication. IF you use any two different kinds of the redox couple, you can have this kind of batteries, but the authors must ensure the impact of the configuration proposed. First and foremost, the point is the Sb one of the excellent alloy type anodes for Na-ion battery, but due to the unavailability, the utilization in practical cells is highly limited. At present, most of the source for Sb is from the recycling (Chem 5 (2019) 3096). The authors proposed the Sb-based system for the grid storage, which requires a huge amount of Sb, I don't know how to fulfill this requirement and claim.

The capacity values given in fractions are very difficult to reproduce; hence, it must be round off. The overpotential/polarization of the new configuration proposed by authors certainly kills the potential use. Therefore, no scope for this kind of cell.

Suppl. Fig. 5 is not properly deconvoluted. This has to be corrected.

To study the nature of the plating/stripping reaction, authors performed CV, in which an extremely high scan rate of 10 mV/s was used. This is not useful in understanding the fundamental reaction.

Extremely poor quality of the XRD pattern was given for the case of NiCo₂O₄

The explanation for the superiority of the FCS over CS is not clear since both the CS and FC composed of the same surface functionalities except its concentration (Fig. 15).

Reviewer #2 (Remarks to the Author):

This work demonstrates a novel Sb anode for aqueous alkaline battery assembly for the first time. The idea of constructing an oxygen-rich interface on carbon substrate to reduce nucleation overpotential of Sb and manipulate diffusion behaviors of the SbO₂⁻ is demonstrated to be simple, interesting and effective. The performance of the as-prepared NiCo//Sb battery is satisfactory. The manuscript is well drafted and the experimental results fully support their concept. Considering the big potential of Sb anode in aqueous battery field, I highly recommend its publication in Nat. Common. after some revisions. My questions are listed below.

1. How to understand the parameter of V_{cl} in Supplementary Fig. 11? For example, the Sb/FCS owns a lower V_{cl} , so what advantage does it have?
2. In Supplementary Fig. 5c, the Y-axis should be "intensity" instead of "intensity (a.u.)". In addition, the atom ratio of the CS and FCS should be given to precisely identify the role of oxygen atoms.
3. The XRD spectra of Sb/CS and Sb/FCS seem to be mistaken. The red line is named as "Sb/CS" and the other one is named as "Sb/FCS", which is not the case for other figures.

4. The total testing time of Sb/CS and Sb/FCS varies widely in Fig. 2. The authors should add extra explanation to clarify why this figure reveals the Sb/FCS presents longer average discharging time and better CE.

5. The authors should point out the mass of two electrodes and how they calculate the energy and power densities.

Reviewer #3 (Remarks to the Author):

The submitted manuscript by zhang et al. demonstrated reversible Sb anodes for aqueous alkaline batteries by constructing an oxygen-rich interface. The reduction of negatively-charged SbO_2^- ions at the negative electrode is facilitated due to adsorption of oxygen functional groups and decreased nucleation barrier. Aqueous alkaline batteries by using Sb as anode and NiCo_2O_4 as cathode exhibit high capacity. Reversible Sb anode is reported for the first time and has potential application for aqueous alkaline batteries. However, there are several points need to be clarified before publication. It is suggested to publish the manuscript on this journal after addressing the major revisions as following:

1. In the Figure 2c, the discharging curves of Sb/CS have obvious potential plateaus at about -0.55 V, which correspond to the stripping of Sb. However, the discharging curves of Sb/FCS have no obvious plateaus while the stripping/plating CV curves of FCS have an anodic peak at -0.42 V. Please explain why the discharging curve of Sb/FCS have no obvious plateaus.
2. In Figure 3b, the deposition on FCS can be divided into two stages: (i) from 0 V to -0.95 V, the potential increases with the capacity; (ii) from -0.95 V to -1.0 V, the deposition curve has a potential plateau. And the capacity at the plateau is only half of the total capacity. Please explain these two processes.
3. The electrolyte is composed of 1 M KOH and 0.027 M $\text{C}_8\text{H}_4\text{K}_2\text{O}_{12}\text{Sb}_2$. Why is the concentration of $\text{C}_8\text{H}_4\text{K}_2\text{O}_{12}\text{Sb}_2$ only 0.027 M? Why did the authors select $\text{C}_8\text{H}_4\text{K}_2\text{O}_{12}\text{Sb}_2$ over other antimonite salts? The salt added in 1 M KOH is $\text{C}_8\text{H}_4\text{K}_2\text{O}_{12}\text{Sb}_2$, while the Sb(III) exists in the form of SbO_2^- . Please give the reaction equation.
4. In 1 M KOH with 0.027 M $\text{C}_8\text{H}_4\text{K}_2\text{O}_{12}\text{Sb}_2$, the potential window of CS and FCS can reach 1.3 V (Figure S6). In this electrolyte, what is the stable window of CS and FCS in 1 M KOH with 0.027 M $\text{C}_8\text{H}_4\text{K}_2\text{O}_{12}\text{Sb}_2$? And what is the overpotential of hydrogen evolution on the CS and FCS electrodes?
5. Does the deposition of antimony have dendrite growth?
6. The oxygen-functional groups can facilitate the surface absorption of SbO_2^- ions in the Helmholtz layer. Would the functional groups promote the charge transfer?
7. Why does the synthesized NiCo_2O_4 need phosphating treatment?
8. The Sb/FCS anode delivers a specific capacity of 627.1 mAh g⁻¹. And NiCo//Sb AAB has a specific capacity of 175.6 mAh g⁻¹. When discussing the capacity (mAh g⁻¹), the mass on which the calculation is based should be clearly demonstrated.
9. The electrochemical curves obtained from three-electrode system in Figure 2 and Figure 3 should give the reference electrode used.

We thank the reviewer for his/her careful review of our manuscript, and really appreciate the constructive comments. Note that all the changes/additions are red-highlighted in the revised version of the manuscript. Please see below for our detailed responses to the comments.

To Reviewer #1:

Authors reported the Sb-based charge storage system, which involves the plating/stripping reaction similar to Zn based system. The work is interesting in the academic aspect and certainly deserves the space. However, the obtained results are not in the level of Nature Communications. Therefore, I am not recommending it for publication. IF you use any two different kinds of the redox couple, you can have this kind of batteries, but the authors must ensure the impact of the configuration proposed.

Response: We sincerely acknowledge your positive evaluation and valuable comments on our work. Yet, we respectfully disagree with your point that any two different kinds of the redox couple can lead to this kind of batteries. In fact, as we mentioned in the introduction section, currently, the limited choice of anodes is the bottleneck for further development of aqueous alkaline batteries (AABs). For those conversion-type anode materials such as Cd, Bi, and FeO_x, they experience phase conversion during the charging/discharging courses (i.e. from metal/metal oxide to their corresponding metal oxide/hydroxide), but the intrinsically poor conductivity and slow reaction kinetics of the metal oxides/hydroxides lead to low capacity and limited rate capability. (*Adv. Mater.* **2018**, *30*, 1707290; *Energy Environ. Sci.* **2017**, *10*, 756-764; *Nat. Commun.* **2012**, *3*, 917) That is the reason why people turn to the stripping/plating-type electrodes relying on the transformation of metal/metal ions for energy storage to achieve fast reaction kinetics. To date, only Zn has been reported as decent stripping/plating-type electrode candidates. However, due to the uncontrollable dendrite growth or side reactions, stripping/plating-type anodes suffer from inferior

cycling stability and low coulombic efficiency (CE). It is thus of great importance to explore novel stripping/plating-type anode materials embedding both high energy and favorable stability to facilitate the practical applications of AABs. The significance of our work lies in that, we, for the first time, successfully demonstrated the feasibility of Sb to function as a stripping/plating-type anode of AABs. Specifically, a smart, facile strategy, the construction of an oxygen-rich functional interface on the substrate to promote the absorption of the SbO_2^- at the interface and to lower the nucleation overpotential of Sb, is developed to induce reversible Sb stripping/plating. Even the energy storage performance of our Sb-based battery is not perfect, considering the interesting and inspiring idea to achieve reversible Sb plating/stripping, as well as its contribution to the development of new-type AABs, we think this work deserves a place in *Nat. Common.*

To highlight the impact of the configuration we proposed, we have carefully revised the manuscript according to your valuable suggestions and additional experiments are also performed to better support our viewpoints. We hope that, in view of the big changes we've made, you could reconsider the publication of our work in *Nat. Common.*

1. First and foremost, the point is the Sb one of the excellent alloy type anodes for Na-ion battery, but due to the unavailability, the utilization in practical cells is highly limited. At present, most of the source for Sb is from the recycling (Chem 5 (2019) 3096). The authors proposed the Sb-based system for the grid storage, which requires a huge amount of Sb, I don't know how to fulfill this requirement and claim.

Response: We fully understand your concern about the relatively low reservation of Sb source. Yet, we still think Sb is a decent anode choice for AABs. First, the mine production of Sb is up to 160 million kg, comparable to many common battery materials such as V (73 million kg), Cd (25 million kg excluding U.S. production) and

Li (77 million kg excluding U.S. production). (*Mineral Commodity Summaries 2020*, U.S. Geological Survey, 2020) Note that these battery systems like V-based flow batteries, Li-metal batteries and Ni-Cd batteries have all been widely used and some of them hold great potential for grid-scale energy storage (*Science* 2020, 367, 30-33; *Chem. Soc. Rev.* 2020, 49, 3040-3071; *Chem. Soc. Rev.* 2018, 47, 8721-8743; *Nat. Rev. Mater.* 2017, 2, 16080; *Nat. Energy* 2019, 4, 180-186; *Nat. Commun.* 2019, 10, 4412; *J. Power Sources.* 2015, 275, 595-604; *Electrochemical Energy Storage for Renewable Sources and Grid Balancing* (pp.223-251), Elsevier B.V., 2015).

Second, the abundance is not the only or most important factor to evaluate the potential of metallic materials. As a proof of concept, nowadays, Li-ion batteries are the most commonly used energy storage devices despite the fact that Li metal does not account for the highest abundance. At present, the recycling of secondary Sb is well developed. The estimated value of secondary Sb produced in the U.S., based on the average New York dealer price for Sb in 2019, was about \$34 million. The bulk of secondary Sb is recovered at secondary lead smelters as antimonial lead, most of which was generated by, and then consumed by, the lead-acid battery industry. Recycling supplies can even hold about 14% of estimated U.S. consumption in 2019 (*Mineral Commodity Summaries 2020*, U.S. Geological Survey, 2020).

Third, for practical application, the cost is one of the most important parameters for reference, which is a comprehensive consideration of reserve, output, mining difficulty, reclamation cost and so on (*Energy Environ. Sci.* 2013, 6, 1083-1092). The low cost of Sb (~ 7\$ per kg) is obviously cheaper than many other metallic anodes like Li (22 \$ per kg, *Energy Environ. Sci.* 2017, 10, 435-459), K (13.1\$ per kg, *CBC Metal* <http://en.cbcie.com/price/13946901.html>) and Bi (17.6-26.4 \$ per kg, *Energy Environ. Sci.* 2017, 10, 435-459). Such distinct cost advantage makes Sb-based batteries strongly competitive in the future.

Above all, the main idea of our work is to demonstrate the feasibility of Sb to function as anode of aqueous devices. We believe that, with the proceeding of basic research,

the abundance problem would be solved through effective recycling or some other advanced strategies. What's more, Sb-based ABBs also see bright future as alternative energy storage devices for some other probable application scenarios such as powering portable device and electric vehicle (*Angew. Chem. Int. Ed.* **2019**, *131*, 14720-14725; *Adv. Mater.* **2017**, *29*, 1700622; *Adv. Energy Mater.* **2020**, *10*, 2000892; *J. Am. Chem. Soc.* **2012**, *134*, 20805-20811; *Energy Storage Mater.* **2020**, *31*, 44-57). To highlight the importance of our work, besides large-scale energy storage, we have elucidated other possible application area of Sb-based aqueous batteries in the revised manuscript.

2. The capacity values given in fractions are very difficult to reproduce; hence, it must be round off.

Response: Thanks for your very good suggestion. In this work, to demonstrate the proper deposition of Sb on CS and FCS, we fixed the charging time to be 55 s and 85 s at 30 mA cm⁻² and 20 mA cm⁻² respectively, and the capacity values given in fractions were actually calculated according to the charging time. To make it easier for the readers to reproduce our experiments, we have clarified the charging time in our revised manuscript.

3. The overpotential/polarization of the new configuration proposed by authors certainly kills the potential use. Therefore, no scope for this kind of cell.

Response: Thanks for your comment. We respectfully disagree with your criticism on the potential use of Sb-based AABs due to its polarization. To disclose the polarization more visually, the CV curves of the NiCo//Sb battery are provided in Fig. R1. Herein, by employing P-NiCo₂O₄ as cathode and Sb/FCS as anode, the obtained NiCo//Sb battery exhibits polarization values of the ~0.5 V at 10 mV s⁻¹, ~0.35 V at 1 mV s⁻¹ and ~0.4 V at 8 mA cm⁻², which is substantially better than or comparable to recently reported AABs at the same test conditions, for example, Ni(OH)₂/MWNT//FeO_x/graphene battery (~0.5 V at 10 mV s⁻¹, *Nat. Commun.* **2012**,

3, 917.); NiCo₂O₄//Bi battery (~0.35 V at 10 mV s⁻¹, *Adv. Mater.* **2016**, *28*, 9188-9195.); Zn/Ni co-doped Co₃O₄@Ni(OH)₂//Fe₂O₃ battery (~0.8 V at 10 mV s⁻¹ and ~0.4 V at 8 mA cm⁻², *Adv. Energy Mater.* **2020**, DOI: 10.1002/aenm.202001064.); NiOOH//3D Zn sponge battery (~0.3 V at 10 mA cm⁻², *Science* **2017**, *356*, 415-418.); Ni(OH)₂/Co(OH)₂//Zn battery (~0.25 V at 10 mV s⁻¹, *Adv. Sci.* **2019**, *6*, 1802002.) and CC-CF@NiO//CC-CF@ZnO battery (~0.48 V at 10 mV s⁻¹, *Adv. Mater.* **2016**, *28*, 8732-8739.). More importantly, our work focuses on the first demonstration of aqueous alkaline battery based on Sb anode, for showing the potential use of Sb stripping/plating chemistry in AAB system. Since this is the first paper related to Sb-based AABs, deep researches in the future are highly expected to optimize the polarization. For instance, the overpotential of Zn-based batteries also went through similar development process. Liu *et al.* reported a NiO//ZnO battery on N-doped carbon cloth-carbon nanofiber in 2016, which had a polarization value of ~0.48 V at 10 mV s⁻¹ (*Adv. Mater.* **2016**, *28*, 8732-8739). After several years, the polarization values of batteries can be reduced to ~0.25 V at 10 mV s⁻¹ after proper modification, by using a 3D Ni@NiO core-shell cathode (*Adv. Funct. Mater.* **2018**, *28*, 1802157) or a nickel-cobalt double hydroxide cathode (*Adv. Sci.* **2019**, *6*, 1802002).

Fig. R1 CV curves under different scan rates of the NiCo//Sb battery.

4. *Suppl. Fig. 5 is not properly deconvoluted. This has to be corrected.*

Response: We are sorry for our mistakes. The XPS figures in Supplementary Fig. 5

have been re-deconvoluted correctly, which can be seen in Fig. R2 and supplementary materials.

Fig. R2 The core level C1s XPS spectra of **a** CS and **b** FCS. **c** The core level O1s XPS spectra of FCS.

5. To study the nature of the plating/stripping reaction, authors performed CV, in which an extremely high scan rate of 10 mV/s was used. This is not useful in understanding the fundamental reaction.

Response: Thanks for your valuable suggestion. We have added CV curves at 2 mV s⁻¹ in Fig. R3 and revised supplementary materials for better understanding the fundamental reaction. Similar to the CV curves at 10 mV s⁻¹, both two electrodes exhibit a redox couple at such a low scan rate, but FCS owns smaller voltage polarization, signifying the better reversibility of Sb plating/stripping courses.

Fig. R3 The stripping/plating CV curves of Sb on **a** CS and **b** FCS under 2 mV s⁻¹.

6. *Extremely poor quality of the XRD pattern was given for the case of NiCo₂O₄.*

Response: Thank you for your suggestion. We have re-performed the XRD measurement of the P-NiCo₂O₄ sample and the XRD pattern has been replaced by the following one now (Fig. R4).

Fig. R4 XRD spectrum of P-NiCo₂O₄ electrode.

7. *The explanation for the superiority of the FCS over CS is not clear since both the CS and FC composed of the same surface functionalities except its concentration (Fig. 15).*

Response: Thanks for your comments. In our work, we do have demonstrated, experimentally and theoretically, that the superiority of the FCS over CS for reversible Sb plating/stripping lies in its higher proportion of oxygen-containing functional groups. Specifically, regular characterizations (e.g. XRD, BET) show FCS and CS have similar crystalline structure and small surface area. However, XPS results reveal that oxygen-containing functional groups in FCS has been dramatically increased compared with that in CS (C/O ratio: from 0.04 to 0.17). The existence of these extra oxygen-containing functional groups facilitates the diffusion and deposition behaviors of SbO₂⁻ ions on the carbon surface via two ways: (i) to promote

the absorption of the SbO_2^- at the interface by activating the formation of hydrogen bonds; (ii) to decrease the deposition resistance of Sb by minimizing the nucleation overpotential on the anode. Since both CS and FCS possess oxygen-containing functional groups, the plating/stripping of Sb is capable of taking place on both substrates even though the reaction efficiency and reversibility are disappointing on the CS. That is, benefiting from the large amount of oxygen-containing functional groups, FCS delivers a more reversible and greater Sb plating.

To Reviewer #2:

This work demonstrates a novel Sb anode for aqueous alkaline battery assembly for the first time. The idea of constructing an oxygen-rich interface on carbon substrate to reduce nucleation overpotential of Sb and manipulate diffusion behaviors of the SbO_2^- is demonstrated to be simple, interesting and effective. The performance of the as-prepared NiCo//Sb battery is satisfactory. The manuscript is well drafted and the experimental results fully support their concept. Considering the big potential of Sb anode in aqueous battery field, I highly recommend its publication in Nat. Commun. after some revisions. My questions are listed below.

Response: Thank you for your positive evaluation and useful comments on this paper. We have carefully revised our manuscript according to your suggestion. The point-to-point responses to your comments can be seen as following:

1. How to understand the parameter of V_{cl} in Supplementary Fig. 11? For example, the Sb/FCS owns a lower V_{cl} , so what advantage does it have?

Response: The V_{cl} represents the energy storage capability of the electrolytic Sb anode in this work. The deposition curve of the anode can be divided in two parts:

efficient deposition part (~ -1.0 V) and HER side reaction (~ -1.5 V). The potential (absolute value) will increase with the charging process. If there are not enough reactant (SbO_2^-) near the electrode surface, the efficient deposition will end prematurely and the potential (absolute value) will increase rapidly, which results in a high V_{cl} . A lower V_{cl} in Sb/FCS means more efficient deposition and better energy storage capability.

2. In Supplementary Fig. 5c, the Y-axis should be “intensity” instead of “intensity (a.u.)”. In addition, the atom ratio of the CS and FCS should be given to precisely identify the role of oxygen atoms.

Response: Thank you for your suggestion. We have corrected the Y-axis and added the corresponding atom ratio in Fig. R5 and Supplementary Fig. 5c.

Fig. R5 The intensity of the O1s peak of two substrates and corresponding O/C atom ratio.

3. The XRD spectra of Sb/CS and Sb/FCS seem to be mistaken. The red line is named as “Sb/CS” and the other one is named as “Sb/FCS”, which is not the case for other figures.

Response: Thanks for pointing out our mistake. The legend is now corrected as

following.

Fig. R6 The XRD spectra of Sb/CS and Sb/FCS.

4. *The total testing time of Sb/CS and Sb/FCS varies widely in Fig. 2. The authors should add extra explanation to clarify why this figure reveals the Sb/FCS presents longer average discharging time and better CE.*

Response: Our charging process is achieved by a galvanostatic charging method to charge a fixed value ($0.47 \text{ mA h cm}^{-2}$) and the discharging process is achieved by a galvanostatic discharging method to discharge to 0 V. The charging time and the cycling numbers of Sb/CS and Sb/FCS are the same. So, the longer test time means longer average discharging time, that is, better coulombic efficiency. Detailed discussion has been added into our revised manuscript.

5. *The authors should point out the mass of two electrodes and how they calculate the energy and power densities.*

Response: Thanks for your kind reminder. The mass loading of Sb/FCS and P-NiCo₂O₄ electrode is 1.1 mg cm^{-2} and 3.1 mg cm^{-2} , respectively. We calculate the volumetric energy and power densities according to the combined volume of cathode

and anode. We have added the relevant information in our revised manuscript and experimental section.

To Reviewer #3:

The submitted manuscript by zhang et al. demonstrated reversible Sb anodes for aqueous alkaline batteries by constructing an oxygen-rich interface. The reduction of negatively-charged SbO_2^- ions at the negative electrode is facilitated due to adsorption of oxygen functional groups and decreased nucleation barrier. Aqueous alkaline batteries by using Sb as anode and NiCo_2O_4 as cathode exhibit high capacity. Reversible Sb anode is reported for the first time and has potential application for aqueous alkaline batteries. However, there are several points need to be clarified before publication. It is suggested to publish the manuscript on this journal after addressing the major revisions as following.

Response: We really appreciate your positive remarks and constructive advices for improving the quality of our work. Accordingly, the manuscript has been carefully revised as you suggest.

1. In the Figure 2c, the discharging curves of Sb/CS have obvious potential plateaus at about -0.55 V, which correspond to the stripping of Sb. However, the discharging curves of Sb/FCS have no obvious plateaus while the stripping/plating CV curves of FCS have an anodic peak at -0.42 V. Please explain why the discharging curve of Sb/FCS have no obvious plateaus.

Response: Thanks for your academic questions. It is worth noting that the test current density is pretty high (20 mA cm^{-2}), so the plateaus of Sb/FCS (~ -0.6 V) is not so obvious in our manuscript. Generally, for the battery test, the lower the current density

is, the more obvious the redox peaks or potential plateaus will be seen. When we measure the discharging curves of Sb/FCS at lower current density (from 2-8 mA cm⁻²), apparent plateaus are detected at -0.6 V on the SB/FCS, as presented in Fig. R7. We have put Fig. R7 in the revised supplementary materials and corresponding explanation is also added.

Fig. R7 Galvanostatic discharging curves at different current densities of Sb/FCS.

2. In Figure 3b, the deposition on FCS can be divided into two stages: (i) from 0 V to -0.95 V, the potential increases with the capacity; (ii) from -0.95 V to -1.0 V, the deposition curve has a potential plateau. And the capacity at the plateau is only half of the total capacity. Please explain these two processes.

Response: Thanks for your good questions. To unravel the deposition behaviors of Sb on FCS substrates, we have followed the weight variation of electrochemically deposited Sb during the two stages. We can see that the electrochemical deposition of Sb does take place on FCS at both stages (0 ~ -0.95 V and -0.95 ~ -1.0 V), as demonstrated by the continuous weight increase of Sb with the potential rise (Fig.R8a). SEM characterization of the electrode at different stages (from I to VI) also well supports this viewpoint (Fig.R8b). That is, the overall capacity of 0.47 mA h cm⁻² is absolutely contributed by the Sb plating on the FCS substrate. In comparison, CS (Fig. R8c) exhibits an increase tendency at the efficient deposition range (0 ~

-1.25 V), and decreased slightly due to the scour of H₂ (-1.25 ~ -1.6 V). Obviously, with respect to the CS, the FCS is more favorable for Sb deposition as its oxygen-rich functional interface can dramatically lower the resistance for Sb nucleation and initiate an underpotential deposition (UPD)-like process of Sb. We have made additional explanation about the two stages in our revised manuscript.

Fig. R8 **a** Weight variation curve and **b** ex-situ SEM images of FCS during the Sb deposition process. **c** Weight variation curve of CS during the Sb deposition process.

3. The electrolyte is composed of 1 M KOH and 0.027 M C₈H₄K₂O₁₂Sb₂. Why is the concentration of C₈H₄K₂O₁₂Sb₂ only 0.027 M? Why did the authors select C₈H₄K₂O₁₂Sb₂ over other antimonic salts? The salt added in 1 M KOH is C₈H₄K₂O₁₂Sb₂, while the Sb(III) exists in the form of SbO₂⁻. Please give the reaction equation.

Response: Thanks for your professional questions. Both Zn and Sb anode are stripping/plating type anode in aqueous alkaline system. Since here we report the first Sb anode for AABs, we chose the concentration of our Sb-based electrolyte referring

to the electrolyte concentration of previously reported Zn-based AABs (*Energy Environ. Sci.* **2014**, *7*, 2025; *Adv. Mater.* **2018**, *30*, 1802396). In addition, compared with other laboratory common Sb(III) salts (such as antimony acetate, antimony triphenyl, and antimony trichloride), antimony potassium tartrate has the advantages of low cost, water solubility and no side-reaction in water. Moreover, the ϕ -pH diagram for H₂O-Sb and experimental results reported in previous works have fully demonstrated that the Sb (III) exists as SbO₂⁻ in alkaline solution (*Chem. Geol.*, **1996**, *130*, 21-30; *J. Sustain. Met.*, **2019**, *5*, 606-616; *Anal. Chim. Acta*, **1998**, *359*, 245-253). The reaction equation for the formation of SbO₂⁻ is listed below:

We have explained why the Sb(III) exists in the form of SbO₂⁻ in the revised manuscript.

4. In 1 M KOH with 0.027 M C₈H₄K₂O₁₂Sb₂, the potential window of CS and FCS can reach 1.3 V (Figure S6). In this electrolyte, what is the stable window of CS and FCS in 1 M KOH with 0.027 M C₈H₄K₂O₁₂Sb₂? And what is the overpotential of hydrogen evolution on the CS and FCS electrodes?

Response: Thank you for the academic questions. As shown in Fig. R9, the overpotential of HER on the CS and FCS are ~435 and ~425 mV, respectively. Both CS and FCS are carbon-based substrates, so they are quite stable in the negative potential interval. However, restrained by the possible electrolyte decomposition, their negative stable potential endpoint in 1 M KOH with 0.027 M C₈H₄K₂O₁₂Sb₂ will be potential where H₂ evolution takes place. That is, the stable potential window for CS is ~-1.51 V and for FCS is ~-1.50 V. As a result, the potential window of 0 to -1.3 V was selected to avoid water splitting.

Fig. R9 Working potential window of the CS and FCS electrodes in 1 M KOH with 0.027 M $C_8H_4K_2O_{12}Sb_2$.

5. Does the deposition of antimony have dendrite growth?

Response: Thanks for your academic questions. For dendrite growth of Sb, we have to say that they may appear at certain conditions but we did not observe any dendrite in this work. It is true that the investigation of the dendrite growth is an interesting topic in the future but this is not the focus of our study here. The novelty of this work is mainly dedicated to the first demonstration of the latent Sb anode application in aqueous alkaline batteries, which heralds new opportunities in the development of aqueous alkaline batteries devices. Further work about probably Sb dendrite growth and its restraint is undergoing.

6. The oxygen-functional groups can facilitate the surface absorption of SbO_2^- ions in the Helmholtz layer. Would the functional groups promote the charge transfer?

Response: Thank you for professional questions. In fact, the oxygen-containing functional groups, especially the O-C=O, will slightly increase the charge transfer resistance. (*Adv. Mater.* **2015**, *27*, 3572–3578) However, these oxygen-containing functional groups can greatly promote the absorption of the SbO_2^- at the interface and

minimize the nucleation overpotential on the anode. Overall, our experiments reveal that the FCS substrate is more favorable for the reversible Sb plating/stripping than CS. We also added electrochemical impedance spectra (EIS) of CS and FCS (Fig. R10) in our revised manuscript and supporting information for more detailed information.

Figure R10 Nyquist plots of CS and FCS.

7. *Why does the synthesized NiCo_2O_4 need phosphating treatment?*

Response: Thanks for your academic question. Our work is mainly focus on the development and optimization of the novel Sb anode, so we use a mature Ni-based material as cathode for demonstration. In fact, this P- NiCo_2O_4 material is based on our previous work about phosphating treatment of NiCo_2O_4 . (*Adv. Mater.* **2018**, *30*, 1802396; *Chem. Eng. J.* **2018**, *352*, 996-1003) Phosphating treatment process can not only increase the active site concentration of NiCo_2O_4 , but also improve its electrical conductivity. More details can be found in our published papers.

8. *The Sb/FCS anode delivers a specific capacity of $627.1 \text{ mA h g}^{-1}$. And NiCo//Sb AAB has a specific capacity of $175.6 \text{ mA h g}^{-1}$. When discussing the capacity (mA h g^{-1}), the mass on which the calculation is based should be clearly demonstrated.*

Response: Anode capacity is based on the mass of deposited Sb, while battery capacity is based on the mass of deposited Sb and as-prepared P- NiCo₂O₄. According to your suggestion, we have added detailed explanations about capacity calculation in revised manuscript.

9. The electrochemical curves obtained from three-electrode system in Figure 2 and Figure 3 should give the reference electrode used.

Response: Thank you for your kind reminder. The reference electrode is saturated calomel electrode (SCE) and we have included this information into our revised figures and revised manuscript.

Reviewer #1 (Remarks to the Author):

I am happy with the revision, hence, I recommend for publication

Reviewer #2 (Remarks to the Author):

The authors have addressed all the comments proposed by the reviewers, and it can be accepted in the present form.

Reviewer #3 (Remarks to the Author):

The authors addressed the questions in the response. The mass loading of the electrode seems low and how to improve it? and the charge mode is based on constant time, which is difficult for the real battery to control the redox reaction state of the electrode.

We thank the reviewer for his/her careful review of our manuscript, and really appreciate the constructive comments. Please see below for our detailed responses to the comments.

To Reviewer #1:

I am happy with the revision, hence, I recommend for publication.

Response: It is our pleasure that you are satisfied with our revisions, and thank you for your valuable efforts in our article review process.

To Reviewer #2:

The authors have addressed all the comments proposed by the reviewers, and it can be accepted in the present form.

Response: Thanks for your efforts in peer review of this paper, which is very helpful to improve the quality of our article.

To Reviewer #3:

The authors addressed the questions in the response. The mass loading of the electrode seems low and how to improve it? and the charge mode is based on constant time, which is difficult for the real battery to control the redox reaction state of the electrode.

Response: We really appreciate your important contributions to our work by peer review. In this work, we used a low mass loading to investigate the stripping/plating behavior of Sb for better comparison, by fixing the charging capacity of $0.47 \text{ mA h cm}^{-2}$. After lengthen the electrodeposition time, the mass loading of the Sb/FCS electrode increased accordingly. For instance, our mass loading of Sb/FCS electrode was increased to 1.1 mg cm^{-2} in NiCo//Sb battery to fulfill the capacity need of the

P-NiCo₂O₄ cathode.

As for charge mode, we used the charge mode of fixed constant time only in single electrode study, for intuitive investigations of some anode characters like over potential and side reactions, which is a general method to study anode materials like Li and Zn (*Science* **2019**, *366*, 645-648; *Nat. Mater.* **2018**, *17*, 543-549). Actually, our charging-discharging test of NiCo//Sb full battery in this work used the charge mode based on potential, just as other batteries do (*Science* **2017**, *356*, 415-418; *Science* **2015**, *350*, 6263).